SCIENCE FORUM

# How failure to falsify in high-volume science contributes to the replication crisis

**Abstract** The number of scientific papers published every year continues to increase, but scientific knowledge is not progressing at the same rate. Here we argue that a greater emphasis on falsification – the direct testing of strong hypotheses – would lead to faster progress by allowing well-specified hypotheses to be eliminated. We describe an example from neuroscience where there has been little work to directly test two prominent but incompatible hypotheses related to traumatic brain injury. Based on this example, we discuss how building strong hypotheses and then setting out to falsify them can bring greater precision to the clinical neurosciences, and argue that this approach could be beneficial to all areas of science.

**SARAH M RAJTMAJER, TIMOTHY M ERRINGTON AND FRANK G HILLARY\***

**\*For correspondence:**
fhillary@psu.edu

**Competing interest:** The authors declare that no competing interests exist.

## Background and motivation

The "replication crisis" in various areas of research has been widely discussed in journals over the past decade [see, for example, *Gilbert et al., 2016*; *Baker, 2016*; *Open Science Collaboration, 2015*; *Munafò et al., 2017*]. At the center of this crisis is the concern that any given scientific result may not be reliable; in this way, the crisis is ultimately a question about the collective confidence we have in our methods and results (*Alipourfard et al., 2012*). The past decade has also witnessed many advances in data science, and "big data" has both contributed to concerns about scientific reliability (*Bollier and Firestone, 2010*; *Calude and Longo, 2017*) and also offered the possibility of improving reliability in some fields (*Rodgers and Shrout, 2018*).

In this article we discuss scientific progress in the clinical neurosciences, and focus on an example related to traumatic brain injury (TBI). Using this example, we argue that the rapid pace of work in this field, coupled with a failure to directly test and eliminate (falsify) hypotheses, has resulted in an expansive literature that lacks the precision necessary to advance science. Instead, we suggest that falsification – where one develops a strong hypothesis, along with methods that can test and refute this hypothesis – should be used more widely by researchers. The strength of a hypothesis refers to how specific and how refutable it is (*Popper, 1963*; see *Table 1* for examples). We also argue for greater emphasis on testing and refuting strong hypotheses through a "team science" framework that allows us to address the heterogeneity in samples and/or methods that makes so many published findings tentative (*Cwiek et al., 2021*; *Bryan et al., 2021*).

### Hyperconnectivity hypothesis in brain connectomics

To provide a specific example for the concerns outlined in this critique, we draw from the literature using resting-state fMRI methods and network analysis (typically graph theory, see *Caeyenberghs et al., 2017* to examine systems-level plasticity in TBI). Beginning with one of the first papers combining functional neuroimaging and graph theory to examine network topology (*Nakamura et al., 2009*), an early observation in the study of TBI was that physical disruption of pathways due to focal and diffuse injury results in regional expansion (increase) in strength or number of functional connections. This initial finding was observed in a small longitudinal sample, but then similar effects were observed in other samples (*Mayer et al., 2011*; *Bharath et al., 2015*; *Hillary et al., 2015*; *Johnson et al., 2012*; *Sharp et al., 2011*; *Iraji et al., 2016*) and animal models of TBI (*Harris et al., 2016*). These findings were summarized

**Table 1.** Examples of hypotheses of different strength.

Exploratory research does not generally involve testing a hypothesis. A Testable Association is a weak hypothesis as it is difficult to refute. A Testable/Falsifiable Position is stronger, and a hypothesis that is Testable/Falsifiable with Alternative Finding is stronger still.

| Type of research/hypothesis | Example |
| --- | --- |
| Exploratory | "We examine the neural correlates of cognitive deficit after brain injury implementing graph theoretical measures of whole brain neural networks" |
| Testable Association | "We hypothesize that graph theoretical measures of whole brain neural networks predict cognitive deficit after brain injury" |
| Testable/Falsifiable Position (*offers possible mechanism and direction/ magnitude of expected finding*) | "We hypothesize that memory deficits during the first 6 months post injury are due to white matter connection loss and maintain a linear and positive relationship with increased global network path length" |
| Testable/Falsifiable with Alternative Finding (*indicates how the hypothesis would and would not be supported*) | "We hypothesize that memory deficits during the first 6 months post injury are due to white matter connection loss and maintain a linear and positive relationship with increased global network path length. Diminished global path length in individuals with greatest memory impairment would challenge this hypothesis" |

in a paper by one of the current authors (FGH) outlining potential mechanisms for hyperconnectivity and its possible long-term consequences, including elevated metabolic demand, abnormal protein aggregation and, ultimately, increased risk for neurodegeneration (see *Hillary and Grafman, 2017*). The "hyperconnectivity response" to neurological insult was proposed as a possible biomarker for injury/recovery in a review summarizing findings in TBI brain connectomics (*Caeyenberghs et al., 2017*).

Nearly simultaneously, other researchers offered a distinct – in fact, nearly the opposite – set of findings. Several studies of moderate to severe brain injury (as examined above) found that white matter disruption during injury resulted in structural and functional disconnection of networks. The authors in these papers outline a "disconnection" hypothesis: the physical degradation of white matter secondary to traumatic axonal injury results in reduced connectivity of brain networks, which is visible both structurally in diffusion imaging studies (*Fagerholm et al., 2015*) and functionally using resting-state fMRI approaches (*Bonnelle et al., 2011*). These findings were summarized in a high-profile review (*Sharp et al., 2014*) where the authors argue that TBI "substantially disrupts [connectivity], and that this disruption predicts cognitive impairment …".

When juxtaposed, these two hypotheses hold distinct explanations for the same phenomenon with the first proposing that axonal injury results in a paradoxically enhanced functional network response and the second that the same pathophysiology results in reduced functional connectivity.

Both cannot be true as they have been proposed, so which is correct? Even with two apparently contradictory hypotheses in place, there has been no direct testing of these positions against one another to determine the scenarios where either may have merit. Instead, each of these hypotheses remained unconditionally intact and served to support distinct sets of outcomes.

The most important point to be made from this example is not that competing theories in this literature exist. To the contrary, having competing theories for understanding a phenomenon places science in a strong position; the theories can be tested against one another to qualify (or even eliminate) one position. The point is that there have been no attempts to falsify either a hyperconnectivity or disconnection hypothesis, allowing researchers to evoke one or the other depending upon the finding for a given dataset (i.e., disconnection due to white matter loss, or functional "compensation" in the case of hyperconnectivity). What has contributed to this problem is that increasingly complex computational modeling also increases the investigator degrees of freedom, both implicitly and explicitly, to support their hypotheses. In the case of the current example of neural networks, these include selection from a number of brain atlases or other methods for brain parcellation and likewise numerous approaches to neural network definition (see *Hallquist and Hillary, 2019*). *Figure 1* provides a schematic representation of two distinct and simultaneously supported hypotheses in head injury.

To be clear, the approach taken by investigators in this TBI literature is consistent with a

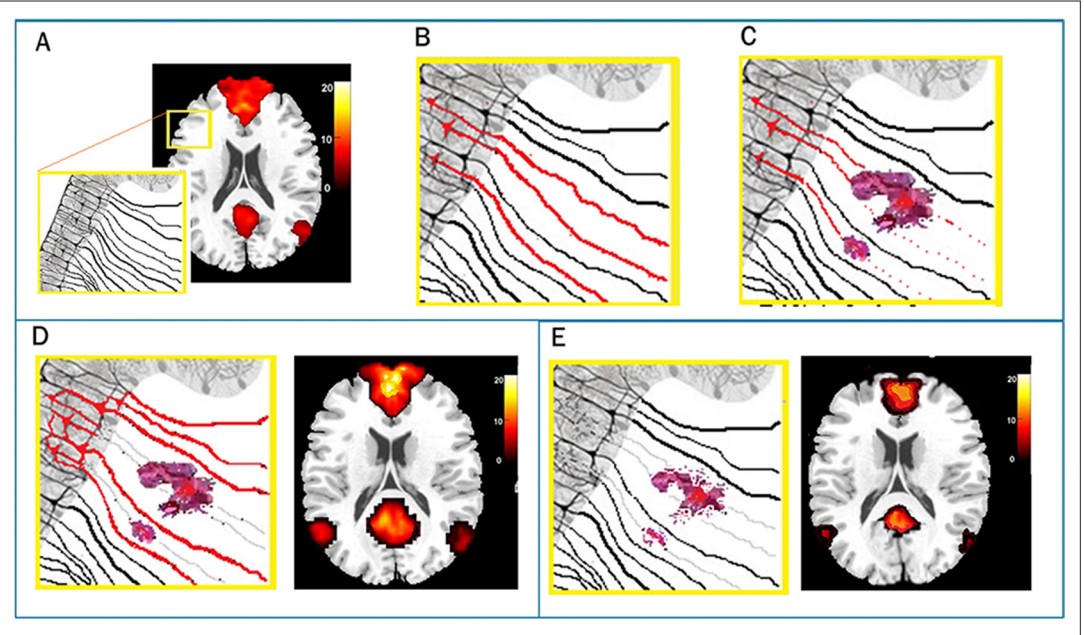

**Figure 1.** Two competing theories for functional network response after brain injury. Panel **A** represents the typical pattern of resting connectivity for the default mode network (DMN) and the yellow box shows a magnified area of neuronal bodies and their axonal projections. Panel **B** reveals three active neuronal projections (red) that are then disrupted by hemorrhagic lesion of white matter (Panel **C**). In response to this injury, a hyperconnectivity response (Panel **D**, left) shows increased signaling to adjacent areas resulting in a pronounced DMN response (Panel **D**, right). By contrast a disconnection hypothesis maintains that signaling from the original neuronal assemblies is diminished due to axonal degradation and neuronal atrophy secondary to cerebral diaschisis (Panel **E**, left) resulting in reduced functional DMN response (Panel **E**, right).

research agenda designed to meet the demands for high publication throughput (more on this below). Examiners publish preliminary findings but remain appropriately tentative in their conclusions given that the sample is small and unexplained factors are numerous. Indeed, a common refrain in many publications is the "need for replication in a larger sample". As opposed to pre-registering and testing strong hypotheses, investigators are reinforced to identify significant results (any result) for publication. In brain injury work examining network plasticity, investigators have often made general claims that brain injury results in "different" or "altered" connectivity (a problem dating back to early fMRI studies in TBI; *Hillary, 2008*). While unintentional, imprecise hypotheses increase the likelihood that chance findings are published. The primary consequence is that all findings are "winners", permitting growing support for either position without movement toward resolution.

Overall, the TBI connectomics literature presents a clear example of a failure to falsify and we argue that it is attributable, at least in part, to the publication of large numbers of papers reporting the results of studies in which small

samples were used to examine under-specified hypotheses. This "science-by-volume" approach is exacerbated by the overuse of inappropriate statistical tests, which increases the probability that spurious findings will be reported as meaningful (*Button et al., 2013*).

The challenges outlined here, where there is a general failure to test and refute strong hypotheses, are not isolated to the TBI literature. Similar issues have been expressed in preclinical studies of stroke (*Corbett et al., 2017*) in the translational neurosciences where investigators maintain flexible theory and predictions to fit methodological limitations (*Macleod et al., 2014*; *Pound and Ritskes-Hoitinga, 2018*; *Henderson et al., 2013*), and in cancer research where only portions of published data sets provide support for hypotheses (*Begley and Ellis, 2012*). These factors have likely contributed to the repeated failure of clinical trials to move from animal models to successful Phase III interventions in clinical neuroscience (*Tolchin et al., 2020*). This example in the neurosciences also mirrors the longstanding problems of co-existing yet inconsistent theories in other disciplines like social psychology (see *Watts, 2017*).

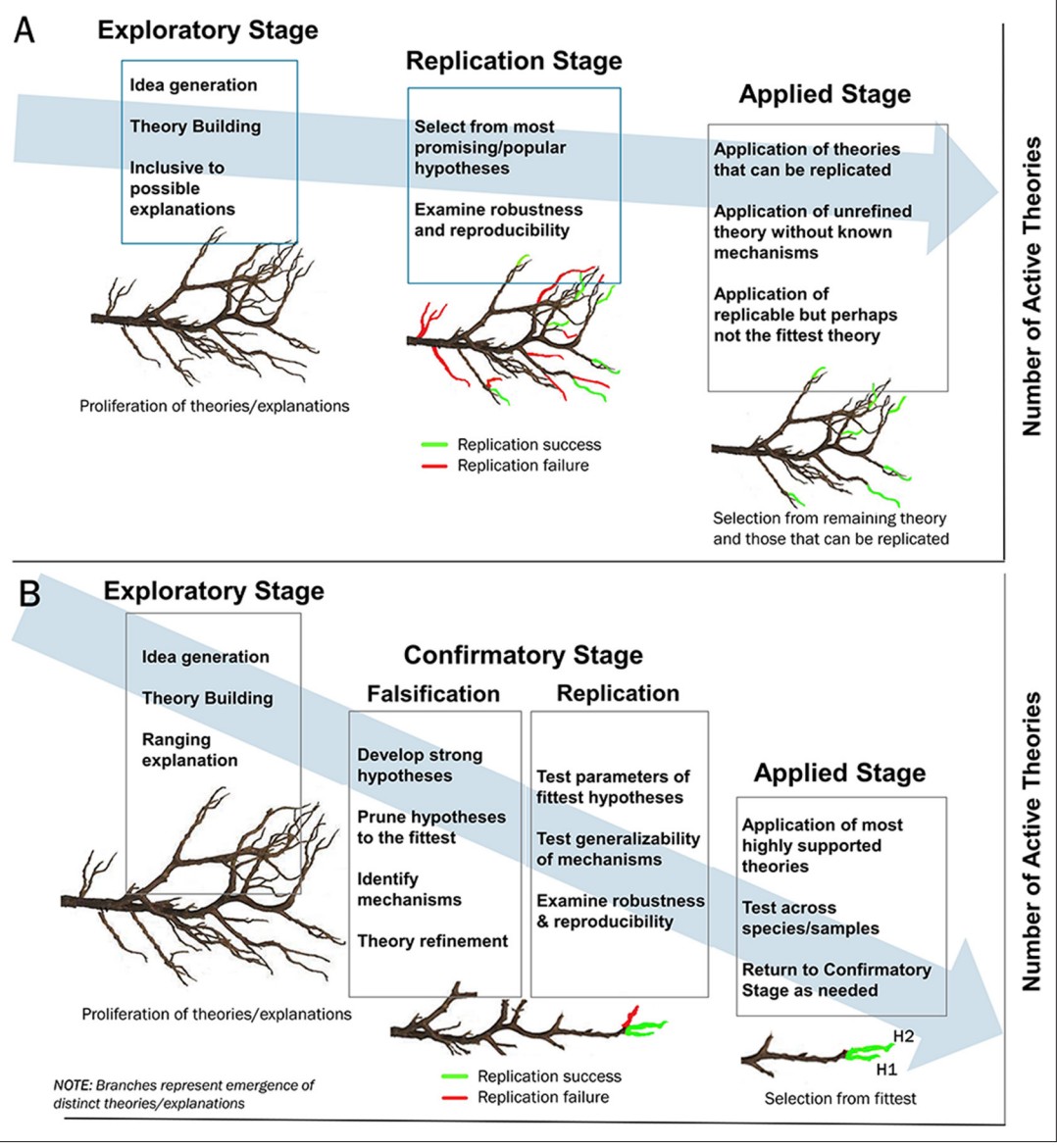

**Figure 2.** The role of falsification in pruning high volume science to identify the fittest theories. Panels **A** and **B** illustrate the conceptual steps in theory progression from exploration through confirmation and finally application. The x-axis is theoretical progression (time) and the y-axis is the number of active theories. Panel **A** depicts progression in the absence of falsification with continued branching of theories in the absence of pruning (theory reduction through falsification). By contrast the "Confirmatory Stage" in Panel **B** includes direct testing and refutation of theories/explanations resulting in only the fittest theories to choose from during application. Note: both Panels **A** and **B** include replication, but falsification during the "confirmation" phase results in a linear pathway and fewer choices from the "fittest" theories at the applied stage.

### Big data and computational methods as friend and foe

The big data revolution and advancement of computational modeling powered by enhanced computing infrastructure, on the one hand, has magnified concerns about scientific reliability through unprecedented flexibility in data exploration and analysis. Sufficiently large datasets provably contain spurious correlations and the number of these coincidental regularities increases as the dataset size increases (*Calude and Longo, 2017*; *Graham and Spencer, 1990*). Adding flexibility, predictive algorithms built on top of these large datasets typically involve a great number of investigator decisions – the combined effects of which undermine reliability of findings [for an example in connectivity modeling see *Hallquist and Hillary, 2019*]. Results of

machine learning models, for example, are sensitive to model specification and parameter tuning (*Pineau, 2021*; *Bouthillier et al., 2019*; *Cwiek et al., 2021*). Computational approaches permit systematically combing through a great number of potential variables of interest and their statistical relationships (specifically, at scales which would be manually infeasible). Consequently, the burden of reliability falls upon the existence of strong, well-founded hypotheses with sufficient power and clear pre-analysis plans. It has even been suggested that null hypothesis significance testing should *only* be used in the neurosciences in support of pre-registered hypotheses based on strong theory (*Szucs and Ioannidis, 2017*).

So, while there is concern that Big Data moves too fast and without the necessary constraints of theory, there is also emerging sentiment that the tremendous computational power coupled with unparalleled data access has the potential to transform some of the most basic scientific tenets, including introduction of a "third scientific pillar" to be added to theory and experimentation (see *National Science Foundation, 2010*). While this latest position received criticism (*Andrews, 2012*), computational methods have been reliably demonstrated to offer novel tools to address the replication crisis – an issue addressed in greater detail in "solutions" below.

### Operating without anchors in a sea of high-volume science

One challenge then is to determine where the bedrock of our field (our foundational knowledge) ends, and where areas of discovery that show promise (but have yet to be established) begin. By some measure neurology is a fledgling field in the biological sciences: the publication of *De humani corporis fabrica* by Vesalius in 1543 is often taken to mark the start of the study of human anatomy (*Vesalius, 1555*) Jean-Martin Charcot – often referred to as the "founder of neurology" – arrived approximately 300 years later (*Zalc, 2018*). If we simplify our task and start with the work of Milner, Geschwind and Luria in the 1950s, it is still a challenge to determine what is definitively known and what remains conjectural in the field. This challenge is amplified by the pressure on researchers to publish or perish (*Macleod et al., 2014*; *Kiai, 2019*; *Lindner et al., 2018*). The number of papers published per year continues to increase without asymptote (*Bornmann and Mutz, 2015*). When considering all papers published in the clinical neurosciences since 1900, more than 50% of the entire literature

has been published in the last 10 years and 35% in the last five years (see supplementary figures S1a,b in *Priestley et al., 2022*). In the most extreme examples, "hyperprolific" lab directors publish a scientific paper roughly every 5 days (*Ioannidis et al., 2018*). It is legitimate to ask if the current proliferation of published findings has been matched by advances in scientific knowledge, or if the rate of publishing is outpacing scientific ingenuity (*Sandström and van den Besselaar, 2016*) and impeding the emergence of new theories (*Chu and Evans, 2021*).

We argue that a culture of science-by-volume is problematic for the reliability of science, primarily when paired with research agendas not designed to test/refute hypotheses. First, without pruning possible explanations through falsification, the science-by-volume approach creates an ever-expanding search space where finite human and financial resources are deployed to maximize breadth in published findings as opposed to depth of understanding (*Figure 2A*). Second, and as an extension of the last point, failure to falsify in a high-volume environment challenges our capacity to know which hypotheses represent foundational theory, which hypotheses are encouraging but require further confirmation, and which hypotheses should be rejected. Finally, in the case of the least-publishable-unit (*Broad, 1981*) a single data set may be carved into several smaller papers resulting in circles of self-citation and the illusion of reliable support for a hypothesis (or hypotheses) (*Gleeson and Biddle, 2000*).

There have even been efforts internationally to make science more deliberate through de-emphasis of publication rates in academic circles (*Dijstelbloem et al., 2013*). Executing this type of systemic change in publication rate poses significant challenges and may ultimately be counterproductive because it fails to acknowledge the advancements in data aggregation and analysis afforded by high performance computing and rapid scientific communication through technology. So, while an argument can be made that our rate of publishing is not commensurate with our scientific progress, a path backward to a lower annual publication rate seems an unlikely solution and ignores the advantages of modernity. Instead, we should work toward establishing scientific foundation by testing and refuting strong hypotheses and these efforts may hold the greatest benefit when used to prune theories to determine the fittest prior to replication (*Figure 2B*). This effort maximizes resources and makes the goals for replication, as a confrontation of theoretical expectations, very clear (*Nosek and Errington, 2020a*). The remainder of the paper

outlines how this can be achieved with focus on several contributors to the replication crisis.

## Accelerating science by falsifying strong hypotheses

### In praise of strong hypotheses

Successful refutation of hypotheses ultimately depends upon a number of factors, not the least of which is the specificity of the hypothesis (*Earp and Trafimow, 2015*). A simple, but well-specified, hypothesis, brings greater leverage to science than a hypothesis that is far reaching with broad implications but cannot be directly tested or refuted. Even Popper wrote about concerns in the behavioral sciences regarding the rather general nature of hypotheses (*Bartley, 1978*), a sentiment that has recently been described as a "crisis" in psychological theory advancement (*Rzhetsky et al., 2015*). As discussed in the TBI connectomics example, hypotheses may have been broad and "exploratory" because authors remained conservative in their claims and conclusions because studies have been systematically under-powered (one report estimating power at 8%; *Button et al., 2013*). While exploration is a vital part of science (*Figure 2*), it must be recognized as scientific exploration as opposed to an empirical test of a hypothesis. Under-developed hypotheses have been argued to be at least one contributor to repeated failure of clinical trials in acute neurological interventions (*Schwamm, 2014*) yet, paradoxically, strong hypotheses may offer increased sensitivity to subtle effects even in small samples (*Lazic, 2018*).

If we appeal to Popper, the strongest hypotheses make "risky predictions", therefore prohibiting alternative explanations (see *Popper, 1963*). Moreover, the strongest hypotheses make clear at the outset the findings that would support the prediction, and also those that would not. Practically speaking this could take the form of teams of scientists developing opposing sets of hypotheses and then agreeing on both the experiments and the outcomes that would falsify one or both positions (what Nosek and Errington refer to as precommitment; *Nosek and Errington, 2020b*). This creates scenarios a priori where strong hypotheses are matched with methods that can provide clear tests. This approach is currently being applied in the "accelerating research on consciousness" programme funded by the Templeton World Charity Foundation. Strong hypotheses must be matched with methods that can provide clear tests, a coupling that cannot be overstated. In the brain

imaging literature alone, there are poignant examples where flawed methods (or misunderstanding of their applications) has resulted in the repeated substantiation of spurious results (in structural covariance analysis see *Carmon et al., 2020* in resting-state fMRI see *Satterthwaite et al., 2012*; *Van Dijk et al., 2012*).

### Addressing heterogeneity to create strong hypotheses

One approach to strengthen hypotheses is to address sample and methodological heterogeneity which plagues the clinical neurosciences (*Benedict and Zivadinov, 2011*; *Bennett et al., 2019*; *Schrag et al., 2019*; *Zucchella et al., 2020*; *Yeates et al., 2019*). To echo a recent review of work in the social sciences, the neurosciences require a "heterogeneity revolution" (*Bryan et al., 2021*). Returning again to the TBI connectomics example, investigators relied upon small datasets heterogeneous with respect to age of injury, time post injury, injury severity, and other factors that could critically influence the response of the neural system to injury. Strong hypotheses determine the influence of sample characteristics by directly modeling the effects of demographic and clinical factors (*Bryan et al., 2021*) as opposed to statistically manipulating the variance accounted for by them – including the widespread and longstanding misapplication of covariance statistics to "equilibrate" groups in case-control designs (*Miller and Chapman, 2001*; *Zinbarg et al., 2010*; *Storandt and Hudson, 1975*). Finally, strong hypotheses leverage the pace of our current science as an ally, where studies designed specifically to address sample heterogeneity can test the role of clinical and demographic predictors in brain plasticity and outcome.

### Open science and sharing to bolster falsification efforts

Addressing sample heterogeneity requires large diverse samples, and one way to achieve this is via data sharing. While data-sharing practices and availability differ across scientific disciplines (*Tedersoo et al., 2021*), there are enormous opportunities for sharing data in the clinical neurosciences (see, for example the Alzheimer's Disease Neuroimaging Initiative (ADNI) and the Transforming Research and Clinical Knowledge in Traumatic Brain Injury (TRACK-TBI) initiative), even in cases where data were not collected with identical methods (such as the Enhancing NeuroImaging Genetics through Meta-Analysis (ENIGMA) Consortium; see *Olsen et al., 2021* for more on severe brain injury, and *Thompson*

*et al., 2020* for a broad summary of work in clinical neuroscience). However, data aggregation and harmonization approaches remain largely untested as a solution to science-by-volume problems in the neurosciences.

It should be stressed that data sharing as a practice is not a panacea to poor study design and/or an absence of theory. The benefits of data combination do not eliminate any existing issues related to instrumentation and data collection occurring at individual sites; it is crucial to understand that data sharing permits faster accumulation of data while retaining any existing methodological concerns (e.g., harmonization). If unaddressed, these concerns introduce magnified noise or systematic bias masquerading as high-powered findings (*Maikusa et al., 2021*). However, well-designed data sharing efforts with rigorous harmonization approaches (e.g., *Fortin et al., 2017*; *Tate et al., 2021*) hold opportunities for falsification through meta-analyses, mega-analyses, and between site data comparisons (*Thompson et al., 2022*). Data sharing and team science also provide realistic opportunities to address sample heterogeneity and site-level idiosyncrasies in method.

Returning to the TBI connectomics example above, data sharing could play a central role in resolving this literature. The neural network response to injury most likely depends upon where one looks (specific neural networks), time post injury, and perhaps a range of clinical and demographic factors such as age of injury, current age, sex, and premorbid status. Clinically and demographically heterogeneous samples of n~40–50 subjects do not have the resolution necessary to determine when hyperconnectivity occurs and when it may give way to disconnection (see *Caeyenberghs et al., 2017*; *Hillary and Grafman, 2017*). Data sharing and team science organized to test strong hypotheses can provide clarity to this literature.

## Harnessing big data to advance metascience

Metascience (*Peterson and Panofsky, 2014*) has become central to many of the issues raised here. Metascience uses the tools of science to describe and evaluate science on a macro scale and to motivate reforms in scientific practice (*Munafò et al., 2017*; *Ioannidis et al., 2015*; *Gurevitch et al., 2018*). The emergence of metascience is at least partially attributable to advances in web search and indexing, network science, natural language processing, and computational modeling. Amongst other aims, work under this umbrella has sought to diagnose biases in research practice (*Larivière et al., 2013*; *Clauset et al., 2015*; *Huang et al., 2020*), understand how researchers select new work to pursue (*Rzhetsky et al., 2015*; *Jia et al., 2020*), identify factors contributing to academic productivity (*Liu et al., 2018*; *Li et al., 2018*; *Pluchino et al., 2019*; *Janosov et al., 2020*), and forecast the emergence of new areas of research (*Prabhakaran et al., 1959*; *Asooja et al., 2016*; *Salatino et al., 2018*; *Chen et al., 2017*; *Krenn and Zeilinger, 2020*; *Behrouzi et al., 2020*).

A newer thread of ongoing efforts within the metascience community is working to build and promote infrastructure for reproducible and transparent scholarly communication (see *Konkol et al., 2020* for a recent review, *Wilkinson et al., 2016*; *Nosek et al., 2015*). As part of this vision, primary deliverables of research processes include machine-readable outputs that can be queried by researchers for meta-analyses and theory development (*Priem, 2013*; *Lakens and DeBruine, 2021*; *Brinckman et al., 2019*). These efforts are coupled with recent major investments in approaches to further automate research synthesis and hypothesis generation. The Big Mechanism program, for example, was set up by the Defense Advanced Research Projects Agency (DARPA) to fund the development of technologies to read the cancer biology literature, extract fragments of causal mechanisms from publications, assemble these mechanisms into executable models, and use these models to explain and predict new findings, and even test these predictions (*Cohen, 2015*).

Lines of research have also emerged using creative assembly of experts (e.g., prediction markets; *Dreber et al., 2015*; *Camerer et al., 2016*; *Camerer et al., 2018*; *Gordon et al., 2020* and AI-driven approaches *Altmejd et al., 2019*; *Pawel and Held, 2020*; *Yang et al., 2020*) to estimate confidence in specific research hypotheses and findings. These too have been facilitated by advances in information extraction, natural language processing, machine learning, and larger training datasets. The DARPA-funded Systematizing Confidence in Open Research and Evidence (SCORE) program, for example, is nearing the end of coordinated three-year long effort to develop technologies to predict and explain replicability, generalizability and robustness of published claims in the social and behavioral sciences literatures (*Alipourfard et al., 2012*). As it continues to advance, the metascience community may serve to revolutionize

the research process resulting in a literature that is readily interrogated and upon which strong hypotheses can be built.

### Falsification for scaffolding convergence research

Advances in computing hold the promise of richer datasets, AI-driven meta-analyses, and even automated hypothesis generation. However, thus far, efforts to harness big data and emerging technologies for falsification and replication have been relatively uncoordinated, with the aforementioned Big Mechanism and SCORE programs amongst a few notable exceptions.

The need to refine theories becomes increasingly apparent when confronted with resource, ethical, and practical constraints that limit what can be further pursued empirically. At the same time, addressing pressing societal needs requires innovation and convergence research. An example are calls for "grand challenges", a family of initiatives focused on tackling daunting unsolved problems with large investments intended to make an applied impact. These targeted investments tend to lead to a proliferation of science; however, these mechanisms could also incorporate processes to refine and interrogate theories as they progress towards addressing a specific and compelling issue. A benefit of incorporating falsification into this pipeline is that it encourages differing points of view, a desired feature of grand challenges (*Helbing, 2012*) and other translational research programs. For example, including clinical researchers in the design of experiments being conducted at the preclinical stage can strengthen the quality of hypotheses before testing them to potentially increase the utility of the result, regardless of the outcome (*Seyhan, 2019*). To realize the full potential, investment in developing and maturing computational models is also needed to leverage the sea of scientific data to help identify the level of confidence in the fitness and replicability of each theory, and where best to deploy resources. This could lead to more rapid theory refinements and greater feedback for what new data to collect than would be possible using hypothesis-driven or data-intensive approaches in isolation (*Peters et al., 2014*).

### Practical challenges to falsification

We have proposed that falsification of strong hypothesis provides a mechanism to increase study reliability. High volume science should ideally function to eliminate possible explanations, otherwise productivity obfuscates progress. But can falsification ultimately achieve this goal? A strict Popperian approach, that every observation represents either a confirmation or refutation of a hypothesis, is challenging to implement in day-to-day scientific practice (*Lakatos, 1970*; *Kuhn, 1970*). What's more, one cannot, with complete certainty, disprove a hypothesis any more than one can hope to prove a hypothesis (see *Lakatos, 1970*). It was Popper who emphasized that *truth* is ephemeral and even when it can be accessed, it remains provisional (*Popper, 1959*).

The philosophical dilemma in establishing the "true" nature of a scientific finding is reflected in the pragmatic challenges facing replication science. Even after an effort to replicate a finding, when investigators are presented with the results and asked if the replication was a success, the outcome is often disagreement resulting in "intellectual gridlock" (*Nosek and Errington, 2020b*). So, if the goal to falsify a hypothesis is both practically and philosophically flawed, why the emphasis? The answer is that, while falsification cannot remove the foibles of human nature, systematic methodological error, and noise from the scientific process, by setting our sights on testing and refuting strong a priori hypotheses we may uncover the shortcomings to our explanations. Attempts to falsify through refutation cannot be definitive but the outcome of multiple efforts can critically inform the direction of a science (*Earp and Trafimow, 2015*) when formally integrated into the scientific process (as depicted in *Figure 2*).

Finally, falsification alone serves as an incomplete response to problems of scientific reliability but becomes a powerful tool when combined with efforts that maximize transparency in method, make null results available, facilitate data/code sharing, and increase the incentive structures for investigators to refocus on open and transparent science.

## Conclusion

Due to several factors including a high-volume science culture and previously unavailable computational resources, the empirical sciences have never been more productive. This unparalleled productivity invites questions about the rigor and direction of science and, ultimately, how these efforts translate to scientific advancement. We have proposed that it should be a primary goal to identify the "ground truths" that can serve as a foundation for more deliberate study and, to do so, there must be greater emphasis on testing and refuting strong hypotheses. The falsification of strong hypotheses enhances the power of replication first by pruning options

and identifying the most promising hypotheses including possible mechanisms. When conducted through a team science framework, the endeavor leverages shared datasets that allow us to address heterogeneity that makes so many findings tentative. We must take steps toward more transparent and open science including – and most importantly – study pre-registration of strong hypotheses. The ultimate goal is to harness the rapid advancements in big data, computational power, and strong, well-defined theory with the goal to accelerate science.

**Sarah M Rajtmajer** is in the College of Information Sciences and Technology, The Pennsylvania State University, University Park, United States

🔗 http://orcid.org/0000-0002-1464-0848

**Timothy M Errington** is at the Center for Open Science, Charlottesville, United States

🔗 http://orcid.org/0000-0002-4959-5143

**Frank G Hillary** is in the Department of Psychology and the Social Life and Engineering Sciences Imaging Center, The Pennsylvania State University, University Park, United States

fhillary@psu.edu

🔗 http://orcid.org/0000-0002-1427-0218

*Author contributions:* Sarah M Rajtmajer, Writing – original draft, Writing – review and editing; Timothy M Errington, Writing – review and editing; Frank G Hillary, Conceptualization, Writing – original draft, Writing – review and editing

*Competing interests:* The authors declare that no competing interests exist.

## Funding

No external funding was received for this work.

### Decision letter and Author response

Decision letter https://doi.org/10.7554/eLife.78830.sa1
Author response https://doi.org/10.7554/eLife.78830.sa2

## Data availability

There are no data associated with this article.

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
