## [Decision Letter]

Thank you for submitting your article "How Failure to Falsify in High Volume Science contributes to the Replication Crisis" to *eLife* for consideration as a Feature Article. Your article has been reviewed by four peer reviewers, and the evaluation has been overseen by a member of the *eLife* Features Team (Peter Rodgers). The following individuals involved in review of your submission have agreed to reveal their identity: Robert Thibault; Jean-Baptiste Poline; Nishant Sinha; and Yujiang Wang.

The reviewers and editors have discussed the reviews and we have drafted this decision letter to help you prepare a revised submission.

Summary:

This manuscript is of broad interest to the scientific community and particularly engaging for the readers in neuroscience. As the number of scientific articles exploring new hypotheses grows exponentially every year, there have not been many attempts to select the most competent hypotheses while falsifying the others. The authors juxtapose two prevalent hypotheses in network neuroscience literature on brain injury as an example. The manuscript offers suggestions on selecting the most compelling hypothesis and directions for designing studies to falsify them. However, there are a number of points that need to be addressed to make the article suitable for publication.

Essential revisions:

1. Lines 55-62: Please give examples of strong and weak hypotheses. Please also explain why "hypotheses that are supported by a range of findings should be considered weak". You might also want to consider linking strong/weak hypotheses to the idea of "research questions", which can be broad and vague, as compared to "hypotheses" which should be more precise.

2. Lines 120-122: Are there any systematic reviews on this topic that use one (or several) strong hypotheses and then check if the various studies support the stronger hypothesis? (e.g., does area X change within timeframe Y, in Z types of patients)? Also, I don't think the issue here is that they lack power to refute hypotheses, but instead that this lack of power is coupled with many statistical tests, leading to false positives that are used to support weak hypotheses (e.g. that brain activity "changes").

3. Lines 230-232: This paragraph could benefit from having a concrete example of a strong hypothesis and a matching weak hypothesis. This will likely help the reader grasp the issue more concretely.

4. Line 261: I am confused by the statement "sample heterogeneity requires large diverse samples". What part of heterogeneity are you trying to address? If testing a treatment that will be deployed to a diverse population of TBIs, I get that you would want a diverse sample, but a diverse sample will increase heterogeneity.

5. Lines 280-282: Or…it might be almost all noise? If we take Button's 2013 finding of 8% power in neuroimaging studies, then differences in findings are more likely due to noise than to demographics or other characteristics.

6. One issue to consider is that the hypotheses considered in the TBI example seem to be too vague or formulated in a too general manner.

7. The interaction between publication incentivization and weak hypotheses is alluded to but leaves the reader unclear on the topic; please say more about this.

8. As noted in the last part of the paper, one of the common tool to limit false discovery is pre-registration. It is unclear why this is not emphasized in the core of the text, rather than lately, as a tool to *review* hypotheses.

9. The social mechanisms to lead us to "team science" are unclear – if this article is meant to help the research community to move in this specific direction, a practical path should be proposed, as "modification of mindset" is certainly more a goal than a practical approach.

10. Specifically, framing all of scientific research in neuroscience in terms of hypotheses that can be confirmed or refuted is limiting; and as the authors acknowledged, the answer might be not a clear yes/no. Please consider discussing how it is sometimes more productive to revise aspects of a particular hypothesis. Nevertheless, I agree with the underlying sentiment that a key to replication is being able to receive recognition for the effort and rigour rather than the outcome.

11. As a computational researcher I would also like to see a stronger emphasis on both:

i) how data, code, and informatics are creating some of the replication crisis;

ii) how stronger informatics frameworks may be part of the solution.

Just an anecdote from personal experience and years of headache: Our lab has been trying to apply a specific networks neuroscience approach called "structural covariance networks". We initially applied it to test the local hyperconnection hypothesis in ASD. It took us years to realise and acknowledge that the computational method itself is very problematic in terms of being replicable even in data from the same scanner, site etc., and despite controlling for every biological variable imaginable. It took us another year, and a very talented student, to understand that it is the method itself that enhances the noise in the data in a very "unuseful" way thus drowning any biological effect. We finally could publish this insight here: https://pubmed.ncbi.nlm.nih.gov/32621973/ I use this example to highlight that our "problem" could not really be framed as a hypothesis refuting exercise, as the real insight for us, and I hope also for the community, was not whether the original hypothesis was right or wrong, but that our tool was flawed.

12. The caption for Figure 2 is confusing, and the content of (Priestley et al., under review) is not clear; please delete this figure and add one or two sentences to the text to say that the number of papers in [subject] has increased from about XX per year in 1970 to YYY per year in 2020.

---

## [Author Response]

Essential revisions:1. Lines 55-62: Please give examples of strong and weak hypotheses. Please also explain why "hypotheses that are supported by a range of findings should be considered weak". You might also want to consider linking strong/weak hypotheses to the idea of "research questions", which can be broad and vague, as compared to "hypotheses" which should be more precise.

This is an important point and based upon this feedback we work to address this issue in text, we now have ~line 57 and we now include a Table with examples.

“In the work of falsification, the more specific and more refutable a hypothesis is, the stronger it is, and hypotheses that can be supported by different sets of findings should be considered weak (Popper, 1963; see Table 1 for example of hypotheses).”

2. Lines 120-122: Are there any systematic reviews on this topic that use one (or several) strong hypotheses and then check if the various studies support the stronger hypothesis? (e.g., does area X change within timeframe Y, in Z types of patients)?

The lack of direct examination of this problem in the hyperconnectivity literature was a primary impetus for the current review. There has been no systematic effort to hold these positions (hyperconnectivity v. disconnection) side-by-side to test them.

Also, I don't think the issue here is that they lack power to refute hypotheses, but instead that this lack of power is coupled with many statistical tests, leading to false positives that are used to support weak hypotheses (e.g. that brain activity "changes").

This is an outstanding point and we agree that the sheer volume of statistical tests in a number of studies increases the probability that findings by chance are published as significant and important. On line 121, we have modified this statement to reflect this point to read:

“Overall, the TBI connectomics literature presents a clear example of a failure to falsify and we argue that it is attributable at least in part by science-by-volume, where small samples are used to examine non-specific hypotheses. This scenario is further worsened using a number of statistical tests which increases the probability that spurious findings are cast as meaningful [40,95].”

3. Lines 230-232: This paragraph could benefit from having a concrete example of a strong hypothesis and a matching weak hypothesis. This will likely help the reader grasp the issue more concretely.

This is an important point and one we discussed as a group prior to the initial submission. We agree completely that concrete examples help to understand the problem and have added several to the text and to Table 1 (line 63).

4. Line 261: I am confused by the statement "sample heterogeneity requires large diverse samples". What part of heterogeneity are you trying to address? If testing a treatment that will be deployed to a diverse population of TBIs, I get that you would want a diverse sample, but a diverse sample will increase heterogeneity.

This is a good point and we have worked to clarify this statement. The point is that small samples do not allow investigation of the effects of sex, education, age, and other factors. Larger, more diverse samples permit direct modeling of these effects. This statement now reads (line 276):

“Addressing sample heterogeneity requires large diverse samples for direct modeling of influencing factors and one avenue to make this possible is data sharing.”

5. Lines 280-282: Or…it might be almost all noise? If we take Button's 2013 finding of 8% power in neuroimaging studies, then differences in findings are more likely due to noise than to demographics or other characteristics.

This is an interesting point, but there does appear to be a there, there. A number of higher powered studies do track changes in connectivity that appear to be directly related to pathophysiology and, importantly, correlate with behavior. However, one cannot deny that at least a subset of these studies presents results that capitalize upon spurious signal or noise.

6. One issue to consider is that the hypotheses considered in the TBI example seem to be too vague or formulated in a too general manner.

The point made here by the Referee is not entirely clear. If this is a statement about the need for stronger hypotheses, we agree that greater specificity is needed and we hope to add context for this point by adding example hypotheses in Table 1. Alternatively, if the Reviewer aims to indicate that this is a TBI-specific phenomenon, that is also possible, though it is unclear why this would occur only in TBI within the clinical neurosciences.

7. The interaction between publication incentivization and weak hypotheses is alluded to but leaves the reader unclear on the topic; please say more about this.

This relationship is made more explicit with the passage on line 115:

“As opposed to pre-registering and testing strong hypotheses, investigators are compelled to identify significant results (any result) for publication. In brain injury work examining network plasticity, investigators have often made general claims that brain injury results in “different” or “altered” connectivity (a problem dating back to early fMRI studies in TBI; [Hillary, 2008]). While it is unlikely the intention, under-specified hypotheses increase the likelihood that chance findings are published. The primary consequence is that all findings are “winners”, permitting growing support for either position without movement toward resolution.”

8. As noted in the last part of the paper, one of the common tool to limit false discovery is pre-registration. It is unclear why this is not emphasized in the core of the text, rather than lately, as a tool to *review* hypotheses.

This is an excellent point, and we now make clear the importance of study preregistration at the outset of the paper (line 55) so that when we return to it, there is context.

9. The social mechanisms to lead us to "team science" are unclear – if this article is meant to help the research community to move in this specific direction, a practical path should be proposed, as "modification of mindset" is certainly more a goal than a practical approach.

We appreciate this point and agree that this statement is confusing. We have removed this statement.

10. Specifically, framing all of scientific research in neuroscience in terms of hypotheses that can be confirmed or refuted is limiting; and as the authors acknowledged, the answer might be not a clear yes/no. Please consider discussing how it is sometimes more productive to revise aspects of a particular hypothesis. Nevertheless, I agree with the underlying sentiment that a key to replication is being able to receive recognition for the effort and rigour rather than the outcome.

This is an important point. Part of the scientific process clearly requires revisions of our theory. As this reviewer alludes to, however, when we revise our hypotheses to fit our outcomes, we risk advancing hypotheses supported by spurious data. We see preregistration as part of the solution and based upon comments elsewhere in this critique, we have refocused on how preregistration can help not only in the development of strong hypotheses, but also in their modification. We also now include Table 1 to provide modern context for what might be considered a “falsifiable” hypothesis. We also include a section titled “Practical Challenges to Falsification” to make clear that falsification of strong hypotheses is one tool of many to improve our science.

11. As a computational researcher I would also like to see a stronger emphasis on both:i) how data, code, and informatics are creating some of the replication crisis;ii) how stronger informatics frameworks may be part of the solution.

We agree and now add a section to outline the parameters of this natural tension (see “Big Data as Friend and Foe”) line 146

Just an anecdote from personal experience and years of headache: Our lab has been trying to apply a specific networks neuroscience approach called "structural covariance networks". We initially applied it to test the local hyperconnection hypothesis in ASD. It took us years to realise and acknowledge that the computational method itself is very problematic in terms of being replicable even in data from the same scanner, site etc., and despite controlling for every biological variable imaginable. It took us another year, and a very talented student, to understand that it is the method itself that enhances the noise in the data in a very "unuseful" way thus drowning any biological effect. We finally could publish this insight here: https://pubmed.ncbi.nlm.nih.gov/32621973/ I use this example to highlight that our "problem" could not really be framed as a hypothesis refuting exercise, as the real insight for us, and I hope also for the community, was not whether the original hypothesis was right or wrong, but that our tool was flawed.

We appreciate the reviewer sharing this illustrative example. It does add a dimension (weak hypothesis v. weak method) that requires recognition in this manuscript. I might additionally argue though that stronger hypothesis (including alternative hypotheses) place the investigator in a better position to detect flawed methodology. That is, truly spurious results may stand-out against sanity checks offered by strong hypotheses, but the point still stands that faulty methods contribute to problems of scientific reliability (something we allude to briefly at the outset with reference to Alipourfard et al., 2021). We now add comment on this on and references to examples for how methods/stats can lead to systematically flawed results (line 238). We now write:

“Strong hypotheses must be matched with methods that can provide clear tests, a coupling that cannot be overstated. In the brain imaging literature alone, there are poignant examples where flawed methods (or misunderstanding of their applications) has resulted in the repeated substantiation of spurious results (in structural covariance analysis see Carmen et al., 2021 and in resting-state fMRI see Satterthwaite et al., 2016; Van Dijk et al., 2012).”

12. The caption for Figure 2 is confusing, and the content of (Priestley et al., under review) is not clear; please delete this figure and add one or two sentences to the text to say that the number of papers in [subject] has increased from about XX per year in 1970 to YYY per year in 2020.

We have accepted this recommendation and have deleted the figure and replaced it with statistics highlighting the annual increase in publication numbers.